# ICAM3-Fc Outperforms Receptor-Specific Antibodies Targeted Nanoparticles to Dendritic Cells for Cross-Presentation

**DOI:** 10.3390/molecules24091825

**Published:** 2019-05-12

**Authors:** Luis J. Cruz, Paul J. Tacken, Johan M.S. van der Schoot, Felix Rueda, Ruurd Torensma, Carl G. Figdor

**Affiliations:** 1Translational Nanobiomaterials and Imaging, Department of Radiology, Leiden University Medical Center, Albinusdreef 2, 2333 ZA Leiden, The Netherlands; 2Department of Tumor Immunology, Radboud Insititute for Molecular Life Sciences, Radboud University Medical Center, Postbox 9101, 6500 HB Nijmegen, The Netherlands; paul.tacken@gmail.com (P.J.T); Bas.vanderSchoot@radboudumc.nl (J.M.S.v.d.S.); Ruurd.Torensma@radboudumc.nl (R.T.); 3Department of Biochemistry and Molecular Biology, University of Barcelona, Diagonal 643, 08028 Barcelona, Spain; frueda@hotmail.es

**Keywords:** ICAM3-Fc, receptor-specific antibodies, targeting, nanoparticles, dendritic cells, cross-presentation

## Abstract

Optimal targeting of nanoparticles (NP) to dendritic cells (DCs) receptors to deliver cancer-specific antigens is key to the efficient induction of anti-tumour immune responses. Poly (lactic-co-glycolic acid) (PLGA) nanoparticles containing tètanus toxoid and gp100 melanoma-associated antigen, toll-like receptor adjuvants were targeted to the DC-SIGN receptor in DCs by specific humanized antibodies or by ICAM3-Fc fusion proteins, which acts as the natural ligand. Despite higher binding and uptake efficacy of anti-DC-SIGN antibody-targeted NP vaccines than ICAM3-Fc ligand, no difference were observed in DC activation markers CD80, CD83, CD86 and CCR7 induced. DCs loaded with NP coated with ICAM3-Fc appeared more potent in activating T cells via cross-presentation than antibody-coated NP vaccines. This fact could be very crucial in the design of new cancer vaccines.

## 1. Introduction

In the last decade, great effort has been put in developing vaccines specifically targeted to dendritic cells (DCs). Such vaccines should generate robust and specific cellular and humoral immune responses [1,2,3] for treatment of cancer or persistent viral infections [4]. DCs are the most efficient antigen-presenting cells (APCs) of the immune system and coordinate innate and adaptive immune responses. DCs are capable of initiating and directing anti-tumour immune responses [5,6,7,8]. Vaccination with soluble cancer antigens (Ags) in non-encapsulated vaccines could be dispersed throughout the body fluids thereby reducing half-life and lowering the efficiency of DCs uptake. Similarly, soluble adjuvants might activate inappropriate (non-antigen presenting) cells resulting in cytokine-related toxicity. This is avoided using Ag and adjuvant loading of DCs cultured “in vitro.” DC based therapies have shown some clinical benefits; however, DC-based vaccine therapies currently in clinical trials involve the ex vivo culturing of monocyte-derived DCs from the peripheral blood of patients, the loading of tumour-specific antigens and adjuvants, followed by the transfer of the cells back into the patient. This is avoided by using DCs generated ex vivo and Ag and adjuvant loaded “in vitro.” However, ex vivo generation of DCs is a complex process, time consuming, costly and requires appropriate GMP facilities, with strict measures necessary for its production. These drawbacks of cellular therapies could be avoided by targeting antigens to DCs in vivo via specific surface receptors using for example poly-(lactic-co-glycolic-acid) (PLGA) particulate delivery systems. PLGA has been successfully used in many papers due to its biodegradability, biocompatibility and versatility. At least 15 PLA/PLGA-based drug products have been approved by FDA on the US market for different drugs and pathologies including chemotherapeutic agents and immune modulators [5,6,7]. PLA/PLGA-based drug products are designed to reduce dosing frequency and potential drug toxicity. PLGA NPs were coated with a lipid-PEG layer to prevent non-specific interactions with cells other than DCs or plasma proteins [8].

Triggering the appropriate receptors might even lead to the activation and maturation of DCs and boosting of immune responses. The choice of drug delivery system will determine the antigen route of entry into the cell and it affects the efficiency of presentation via the MHC class I and II pathways. One of the greatest benefits of particle-based antigen delivery systems resides in their capacity to carry Ag and adjuvants concomitantly to the same APC, which is crucial for efficient induction of immune responses [4,9,10,11]. Furthermore, the targeted delivery of Ag to DC surface receptors enhances presentation to T cells [12].

Initially, DC-targeted vaccines were generated by fusing antigens to antibodies recognizing C-type lectin receptors specifically expressed by DCs. Recently, we developed a nanoparticle (NP) system that carried multiple vaccine components to DCs [7,8]. Targeted delivery of these NP not only improved antigen presentation but also increased the adjuvanticity of TLR ligands (TLRLs). Simultaneous targeted delivery of multiple vaccine components can be accomplished by encapsulation within biodegradable NP vaccine carriers harbouring antibodies (Abs) that recognize DC surface receptors [8,13,14,15,16,17,18,19,20,21].

Several members of the C-type lectin receptor (CLR) family are mainly expressed by DCs, which allow the target of antigens directly to DCs in vivo [4]. When DC maturation stimuli are co-administered with antigens targeted through CLRs induce antigen presentation and activate the immune response [22]. In general, vaccines currently on the market mainly induce humoral responses; however, some DC subsets, when targeted to the appropriated receptors, also induce strong cytotoxic T cell (CTL) responses [4] due to their ability to present exogenous antigens through MHC class I, a process known as cross-presentation. 

The CLRs has a carbohydrate recognition domain (CRD) which binds carbohydrate structures, like specific mannose, galactose or fucose present on self or non–self-proteins in a Ca^2+^-dependent manner [23].

In this work DC-SIGN (dendritic cell-specific intercellular adhesion molecule-3 (ICAM3)-grabbing non-integrin), a member of the CLR family previously explored, has been assayed. DC-SIGN present an extracellular part composed of a C-terminal CRD and a neck region consisting several 23-residue tandem repeat regions, which recognize and internalize many pathogens, including viruses, bacteria, fungi and parasites [24]. Then the pathogen-derived antigens are presented via MHC class I and II molecules to CD8+ and CD4+ T cells, respectively [25,26]. This natural phenomenon is used as vaccination strategy to targeted antigens to the CRD of DC-SIGN and by analogy they will be presented via MHC classes I and II [27].

Although the precise mechanism that triggers receptor internalization is still unknown, both antibodies and sugar ligands have been used to target antigens to the CRD of DC-SIGN [28,29]. We previously demonstrated that internalization of the receptor when the ligand binds to the CRD of DC-SIGN is a clathrin-mediated process and once inside the cell they go to late endosomal, LAMP-1 and MHC class II positive, compartments [30,31]. Vaccination strategies employ either carbohydrates or specific Abs to target DC-SIGN. In general, Abs bind their target with high affinity and specificity, whereas carbohydrates bind with lower affinities and are recognized by multiple receptors [32,33]. Therefore, it is difficult to determine which receptor or receptors are responsible in vivo for the immune responses induced by vaccination with carbohydrate-targeted vaccines/Ags [18]. However, at least *in vitro*, carbohydrate-modified Ags bind effectively to human DC-SIGN on DCs, resulting in Ag presentation via MHC classes I and II [34]. Specificity and affinity for DC-SIGN may not be the only two factors determining the immunological outcome of targeted vaccines.

Several Abs and carbohydrates (such as Lewis-X (LeX), mannosylated lipoarabinomannan (ManLAM) and carbohydrate moiety of gp120) were reported to activate signalling pathways upon DC-SIGN ligation [18,34]. Depending on the nature of the ligand, DC-SIGN signalling enhances or suppresses proinflammatory responses [9,10]. The extent to which these signalling pathways affect Ag presentation and T cell activation has not been the subject of detailed studies.

ICAM3 is a natural ligand of DC-SIGN on DCs and is highly expressed on the T cell surface. DC-SIGN binds ICAM3 with high affinity and is important for the first contact between DCs and T cells that support the primary immune response [35]. ICAM-3 can also bind to LFA. However, LFA has to be activated to allow binding. A resting T cell expresses low affinity LFA and upon activation LFA changes its conformation and forms clusters on the membrane providing high affinity to ICAMs. Here we compared NP vaccines carrying clinically relevant Ags and adjuvants that were coated with two distinct DC-SIGN ligands: a specific humanized antibody engineered not to interact with Fc receptors [36] and composed fusion protein of ICAM-3 and the Fc moiety of human IgG1 [37]. We compared both ligands of the PLGA-NP targeted vaccines with regard to their potential to activate primary human DCs and subsequent activation of human T cells, due to the importance in the design of DC-SIGN targeted vaccines for effective clinical application.

Despite of a stronger affinity of the antibodies for DC-SIGN, ICAM3-Fc induced more efficient cross-presentation of Ag by DCs to T cells.

## 2. Results

### 2.1. Characterization of NP Vaccine

NP vaccines were generated using the biodegradable polymer poly(lactic-co-glycolic acid) (PLGA). Figure 1 shows a schematic diagram of the DC-SIGN-targeted NP vaccine. The encapsulation efficiency of peptides and TLRLs within the carriers was determined by reverse phase high-performance liquid chromatography. Overall, we found that the encapsulation efficiency of the TT and gp100 peptide Ag was superior to that of the TLRLs (Table 1). As shown in Table 2, the diameter of the PLGA-based carriers, as determined by dynamic light scattering (DLS), was approximately 200 nm.

Carriers were coated with a lipid-PEG layer to prevent non-specific interactions with cells or proteins [8]. Streptavidin was covalently linked to the lipid-PEG layer and biotinylated DC-SIGN ligands were introduced: the ICAM3-Fc and anti-human DCSIGN antibodies hD1 (αDC-SIGN) and isotype control antibodies, as negative controls.

Introduction of ICAM3-Fc and antibodies resulted in a slight increase in carrier size. The coating efficiency was higher when using the relatively small protein (±80 kDa) than when using the antibody (±150 kDa) (Table 2).

### 2.2. Binding and Uptake on DCs of NP Vaccine Coated with DC-SIGN Ligands

NP vaccine harbouring FITC-labelled TT peptide or gp100 Ags and coated with DC-SIGN ligands were assayed for binding at 4 °C to stop uptake transport and uptake at 37 °C on DCs by flow cytometry (Figure 2). Binding to and uptake by, DCs was more efficient for DC-SIGN specific antibody-coated (29% and 50% respectively) than ICAM3-Fc-coated NP vaccines (17% and 26% respectively) (Figure 2A,E). Binding and uptake of NP coated with DC-SIGN specific antibodies or ICAM3-Fc was strongly reduced by pre-blocking DC-SIGN with receptor-specific antibodies, as expected (Figure 2B,F). In contrast, blocking Fc receptors by addition of excess IgG hardly affected binding and uptake of NP (about 25%) (Figure 2C,D,G,H). This indicates that ICAM3-Fc coated NP are targeted to DCs via the ICAM3 moiety of the fusion protein and to a much lesser, but not negligible extent, by the Fc moiety; acting together, they might lead to very efficient uptake and cross-presentation. The DC-SIGN-specific antibody was engineered with a hybrid IgG2/IgG4 Fc moiety and is therefore unable to interact with Fc receptors [36]. Some residual uptake in the negative controls was observed (isotype and non-coated), mainly at 37 °C. This indicates the DCs phagocytosed NP despite the PEG coating, although uptake was low.

### 2.3. NP Vaccines Targeted by Anti-DC-SIGN and ICAM3-Fc Activate DCs

The importance of incorporating TLR ligands as adjuvant in the NP vaccines was revealed by the inability of NP carrying only peptide to induce DCs maturation, as determined by analysing surface markers of maturation, such as CD80, CD83, CD86 and CCR7 (Figure 3A). Incorporation of TLR ligands in NP vaccines resulted in potent DC maturation but only when particles were targeted to DCs by the DC-SIGN antibody or ICAM3-Fc (Figure 3B). Despite anti-DC-SIGN antibody was more efficient in target NP to DCs than ICAM3-Fc (Figure 2A,E), this did not result in superior DC maturation. This fact correlated with the MHC class II-restricted recall T cell proliferation assay, where more proliferation was observed when NP carried TLR ligands but no differences were observed between targeting with ICAM3-Fc or DCSIGN-specific antibody (Figure 3C,D).

Production of IL-6, IL-10 and TNF-α upon treatment of DC with NP vaccines containing TLR ligands was similar for antibody- and ICAM3-Fc-targeted NPs. NPs targeted by anti-DC-SIGN antibody showed a trend of enhancing IL-8 and IL-12 production when compared to NPs targeted by ICAM3-Fc (Figure 3E). However, the difference was not significant. This may reflect their increased binding to and uptake by DCs, as shown in the binding and uptake assays (Figure 2).

### 2.4. ICAM3-Fc Coating NP Vaccine Induces the Highest Levels of Cross-Presentation

Previously, anti-DC-SIGN antibodies were shown to be more efficient in inducing cross-presentation when compared to many other DC-SIGN ligands [18]. Despite the fact that ICAM3-Fc targeted DCs less efficiently (Figure 2) and induced comparable levels CD4+ T cell activation (Figure 3C,D) as compared to Abs directed against DC-SIGN, it strongly enhanced activation of CD8+ T cells, as measured by the early activation marker CD69 (Figure 4A). Moreover, ICAM3-Fc coated NP vaccines resulted in enhanced IFN-γ production by CD8+ T cells compared to Ab-coated NP vaccines (Figure 4B).

These results indicated that, despite the fact that DC-SIGN antibody results in more efficient targeting (Figure 2), ICAM3-Fc fusion protein was superior in inducing Ag cross-presentation.

This is striking as DC-SIGN-targeted antibodies outperformed all other carbohydrate DC-SIGN ligands (Lewis-X, mannosylated lipoarabinomannan and glycosylated HIV protein gp120) tested in our previous studies with respect to induction of CD8+ T cell activation [18].

### 2.5. NP Vaccines Coated with ICAM3-Fc Induce Cross-Presentation via DC-SIGN but does not Require FcRs CD32 or CD64

To determine whether Ag presentation by ICAM3-Fc targeted NP vaccines required ICAM3 binding to DC-SIGN, Fc binding to Fc receptors or ICAM3 binding to its ligand LFA-1 (expressed on T cells) we performed blocking experiments. Activation of CD4+ and CD8+ T cells by tetanus toxoid (TT) peptide and gp100 (272–300) peptide by DCs loaded with ICAM3-Fc targeted NP vaccine was abrogated by blocking DC-SIGN by DC-SIGN-specific antibodies or EDTA. Binding of DC-SIGN to carbohydrates requires calcium, which is unavailable when EDTA is present. T cell activation was also impaired by an FcR blocking reagent, consisting of a high concentration of human IgG.

Specific antibodies against LFA-1, the ICAM3 ligand on T cells, could not impair T cell activation (Figure 5A,B), possibly compensated for by other mechanism. Under the same conditions, IFN-γ production by CD8+ T cells was blocked in the presence of the FcR blocking reagent and DC-SIGN blocking antibodies, while blocking of LFA-1 had no effect (Figure 5C). These results indicate that ICAM3-Fc NP vaccines are targeted to DCSIGN and require both ICAM3 and Fc function. The Fc moiety of ICAM3-Fc could bind to Fc receptors or other receptors. Surprisingly, DC-SIGN itself was recently reported to interact with IgG [38,39,40].

In order to confirm whether or not ICAM3-Fc targeted Fc receptors, experiments were carried out blocking CD32 and CD64 (which are the Fc receptors most prominently expressed by monocyte-derived DC) with specific Abs. Neither antibody was able to block cross-presentation induced by ICAM3-Fc targeted NP vaccine (Figure 5D).

However, mouse IgG1 (mIgG1, isotype control for CD32 and CD64 Abs) was able to appreciably inhibit activation induced by ICAM3-Fc, suggesting mouse IgG1 competed with the human Fc portion of ICAM3-Fc for binding to another receptor [40,41] which may be the FcγRIIB, the only inhibitory IgG receptor. DCs activated with excess soluble TLRL and pulsed with the minimal gp100 peptide (gp100, 280–288) yielded maximum CD69+ T cell activation of these TCR-transfected T cells. The fact that 65% of all T cells were activated may reflect that not all T cells were transfected with the gp100-specific TCR.

### 2.6. Adding FcR Binding Capacity to Antibody Coated NP Vaccine does not Improve Cross-Presentation

The DC-SIGN targeting Ab used in this study (clone hD1 G2/G4) does not interact with Fc receptors, because it contains a hybrid IgG2/IgG4 Fc tail [36]. Not surprisingly, blocking Fc does not affect cross-presentation of NP vaccines targeted by hD1 G2/G4 (Figure 6A). The fact that the Fc moiety of ICAM3-Fc is partly responsible for superior cross-presentation suggests targeting Abs may benefit from a functional Fc tail. To test this hypothesis, we performed cross-presentation experiments with the Ab targeted NP vaccines in the presence of functional Fc. Cross-presentation by DCs incubated with hD1 G2/G4 targeted NP vaccines was not enhanced by addition of Fc-containing immune complexes, which may enhance DC activation by FcR triggering.

Immune complexes were generated by incubation of the protein Ag KLH with serum of individuals previously immunized with and containing antibodies directed against, KLH [36]. Pre-immune serum of the same individuals incubated with KLH served as a negative control. Coating of NP vaccines with both hD1 G2/G4 and IgG Fc protein (molar ratio 1:1) to generate particles that may simultaneously bind DC-SIGN and Fc receptors did also not enhance cross-presentation to coating with hD1 G2/G4 alone.

Moreover, NP vaccine targeted to Fc receptors by coating with IgG Fc protein alone induced comparable levels of cross-presentation to those coated with hD1 G2/G4 alone (Figure 6A). Next, we replaced the hybrid G2/G4 Fc of the hD1 G2/G4 Ab by a regular human IgG1 Fc (hD1 G1) to generate a targeting moiety that combines an IgG1 Fc tail and DC-SIGN binding within a single molecule, similar to ICAM3-Fc. Targeted delivery of NP vaccine via the hD1 G1 Ab induced similar levels of DC cross-presentation as observed by targeted delivery through hD1 G2/G4 (Figure 6B). In contrast to the situation with ICAM3-Fc targeted NP vaccines, Fc block did not impair cross-presentation induced by hD1 G1 targeted NP vaccine (Figure 5D and Figure 6B). Blocking Fc receptors CD32 and CD64 had no effect on DC cross-presentation induced by hD1 G1 targeted NP vaccines antibodies directed against the Fc receptors CD32 and CD64 (Figure 6B). Together the results show that cross-presentation induced by Ab-mediated targeting of NP vaccine to DC-SIGN is not enhanced by simultaneous triggering of Fc receptors. The experiments further support the idea that NP vaccines targeted via ICAM3-Fc induce efficient CD8+ T cell activation by an Fc receptor-independent pathway, despite the fact that the Fc moiety is indispensable for this effect.

## 3. Discussion

Targeting biodegradable NP vaccine carrying Ags and adjuvant molecules to specific receptors on DCs is a very promising strategy to develop therapeutic vaccines against cancer and persistent virus infections. A plethora of techniques has been developed in the last decades that enabled the discovery of many viral Ags and cancer-associated Ags with importance for the generation of therapeutic vaccines. Also, adjuvants, like the TLR ligands, capable of inducing a strong immune response and even reversing or at least modulate, the anergic state of T cells present in the tumours of patients with advanced stages of cancer, have been discovered [42]. However, how NP vaccines carrying Ags and adjuvants are best targeted to DCs to most effectively activate DCs to induce anti-tumour T cell responses is still the subject of intense research. DCs have a large number of receptors that can be targeted, of which some induce phagocytosis, boost, stimulate DCs activation and/or Ag cross-presentation. DC-SIGN and Fc receptors both constitute receptors showing all three properties.

In this study, a fusion protein consisting of ICAM3, a natural ligand of DC-SIGN, linked to the Fc fraction of the human IgG1, was compared to a specific humanized antibody directed to DC-SIGN (hD1 G2/G4). Surprisingly, despite the fact that NP vaccine targeted by DC-SIGN specific Ab was more efficiently binding to and taken up by, human DCs, DCs incubated with NP vaccine targeted by ICAM3-Fc induced unprecedented levels of CD8+ T cell activation by Ag cross-presentation. The percentage of CD8+ T cells activated by DCs incubated with ICAM3-Fc NP vaccine carrying the long gp100 peptide was almost as high as that of DCs matured with excess soluble TLR ligands and exogenously pulsed with saturating amounts of the minimal gp100_280-88_ peptide epitope (Figure 5D). The effect on CD4+ T cell activation was much less pronounced: DCs incubated with ICAM3-Fc or hD1 G2/G4 targeted NP vaccines were equally efficient in activating CD4+ T cells via an MHC class-II restricted peptide. We tested the hypothesis that ICAM3-Fc enhances cross-presentation of DCs by simultaneously engaging DC-SIGN and Fc receptors, since it is well known that Fc receptors activate DCs by means of interactions of immune complexes or Fc moiety of IgG1 [42,43,44,45,46]. The data suggested that both DC-SIGN binding and the IgG1 Fc moiety of ICAM3-Fc were indispensable for efficient CD8+ T cell activation. However, blocking of Fc receptors CD32 and CD64 on DCs did not reduce cross-presentation. Neither did addition of the IgG1 moiety of ICAM3-Fc to the targeting DC-SIGN Ab result in a targeting Ab that could induce CD8+ T cell activation to the same level as ICAM3-Fc.

This suggests that combined DC-SIGN binding and Fc triggering is not sufficient to induce potent Ag cross-presentation resulting in CD8+ T cell activation. Besides the DC-SIGN engaging ability of ICAM3-Fc and the requirement of the IgG1 Fc moiety, ICAM3-Fc must contain other properties that render it extremely potent for inducing MHC class I-restricted responses. Experiments wherein ICAM3-Fc coated NP vaccines were added to DCs and T cells from different donors did not enhance the mixed lymphocyte reaction, indicating ICAM3-Fc NP vaccines do not enhance CD8+ T cell activation by direct Ag-independent action on DCs or T cells (data not shown, LC).

Most likely, ICAM3-Fc activates other receptors which enhance the cross-presentation of the encapsulated tumour Ag by the DC.

ICAM3-Fc binds DC-SIGN via carbohydrate residues present on ICAM3 and binding may be blocked by addition of human serum IgG. This could be justified, because IgG also contains carbohydrate residues that may interact with DC-SIGN [38,39,40]. It is not clear why the ICAM3-Fc sugars then are more efficient in mediating cross-presentation than DC-SIGN antibodies or other DC-SIGN binding sugar residues. It is tempting to speculate that they induce a specific signalling cascade via DC-SIGN or interacting in cytoplasm with other molecules, thereby increasing cross-presentation or that the ICAM3-Fc sugars bind to DC-SIGN and also to a so far unknown additional receptor.

Further research is required to unravel such underlying mechanisms, as it teaches us how vaccines can be developed that induce potent cellular immunity to treat cancer and persistent viral infections.

## 4. Materials and Methods

### 4.1. Materials

PLGA (Resomer RG 502 H, lactide/glycolide molar ratio 48:52 to 52:48) was purchased from Boehringer Ingelheim (Bonn, Germany). Solvents for peptide synthesis and PLGA preparation (dichloromethane, N,N’-dimethylformamide and ethyl acetate) were obtained from Merck (Ingelheim am Rhein, Germany). EDAC (1-ethyl-3-[3-dimethylaminopropyl] carbodiimide), NHS (N-hydroxysuccinimide) and sulfo-NHS-LC-Biotin were obtained from Pierce (Landsmeer, The Netherlands). Polyvinyl alcohol (PVA) was purchased from Sigma (Zwijndrecht, Netherlands). R848 was from Axorra (Farmingdale, NY, USA), poly I:C and streptavidin from Sigma. The following antihuman antibodies were used: anti-human HLA-DR/DP clone Q5/13, CD83 (Beckman Coulter, Woerden, Netherlands) and CD69, CD80, CD86 and CCR7 (all from BD Biosciences, Vianen, The Netherlands). Alexa Fluor 647-labeled goat anti-human and goat anti-mouse IgG (Invitrogen, Carlsbad, CA, USA). The previously described humanized anti-DC-SIGN (hD1) and its isotype control (h5G1.1) were kindly provided by Alexion Pharmaceuticals (Boston, MA, USA) [36,38]. ICAM3-Fc recombinant protein was produced in CHO K1 cell line (ATCC. LGC Standards GmbH. Wesel, Germany) and purified using protein G column according to manufacturer protocol (GE Healthcare, Chicago Il, USA). After purification, the recombinant protein was characterized via SDS-PAGE and ELISA. Antibodies and ICAM3-Fc protein were biotinylated using Sulfo-NHS-LC-Biotin according from Pierce (Landsmeer,The Netherlands) manufacture protocol [37].

### 4.2. Cells

Leukocytes were obtained from buffy coats of healthy individuals and purified using Ficoll density centrifugation. Monocyte-derived DCs were obtained from leukocytes as reported elsewhere [36]. Cells were cultured in X-VIVO 15 medium (Cambrex, Wiesbaden, Germany) supplemented with 2% human serum. CD8+ T cells transfected with a TCR recognizing the gp100 (280–288) epitope were generated as described previously [39]. Blood cells from controls were obtained from buffy coats from anonymous healthy donors of the blood bank after informed consent. Experiments were approved and carried out in accordance with the guidelines and regulations of the Radboud University Medical Centre, Nijmegen, The Netherlands.

### 4.3. Peptide Synthesis

The gp100 (272–300) (RALVVTHTYLEPGPVTAQVVLQAAIPLTS) long peptide and the tetanus toxoid (TT) epitope that comprises the 830–844 region of the protein containing fluorescein isothiocyanate (FITC) (FITC-**KK**QYIKANSKFIGITEL**KK**-COOH) at the N-terminal were synthesized manually according to standard protocols for solid-phase peptide synthesis, using the Fmoc/tert-butyl strategy [47].

### 4.4. Generation of Targeted NP Vaccine

NP vaccine carriers coated with streptavidin and carrying ICAM3-Fc or DC-SIGN antibody were generated using the copolymer poly(lactic-co-glycolic acid) using an o/w emulsion and solvent evaporation–extraction method. In brief, 50 mg of PLGA in 3 mL of methylene chloride containing 1) FITC-TT peptide (2 mg in 100 µL in water) or 2) DQ-BSA (2 mg in 100 µL in water) was added dropwise to 25 mL of aqueous 2% PVA and emulsified for 90 s using a sonicator (Branson, sonofier 250). A combination of 6 mg DSPE-PEG(2000) maleimide and 6 mg mPEG-2000 PE were dissolved in methylene chloride and added to the vial.

The methylene chloride was removed by a stream of nitrogen gas. Subsequently, the emulsion was rapidly added to the vial containing the lipids and the solution was homogenized for 30 s using a sonicator. Following overnight evaporation of the solvent at 4 °C, the NPs were collected by ultracentrifugation at 60,000× *g* for 30 min, washed three times with distillated water and lyophilized, essentially as described before [13,41]. Peptides were encapsulated by adding gp100 (272–300) (2 mg) or FITC-TT-peptide (2 mg) and TLRLs were encapsulated by adding poly I:C (4 mg in 100 µL PBS) and R848 (0.8 mg in 50 µL H2O/DMSO, 9:1) to 50 mg of PLGA.

The encapsulation efficiency of peptides, poly I:C and R848 was determined by reverse phase high-performance liquid chromatography (RP-HPLC) [41]. Streptavidin (2 mg) was covalently coupled to 10 mg of NP vaccine carriers by activating (PLGA-PEG-COOH) surface carboxyl groups in isotonic 0.1 M MES saline buffer pH 5.5 containing EDAC (10 equiv.) and NHS (10 equiv.) for 1 h. Next, NP vaccine carriers were washed in PBS and reacted with streptavidin (2 mg). The coated NP were then centrifuged and washed with PBS buffer to remove any unbound streptavidin. Next, the biotinylated DC-SIGN ligands (biotin-antibodies and Biotin-ICAM3-Fc) were added to the carriers for 1 h at 4 °C.

Excess ligands were removed by centrifugation at 25,000× *g* for 30 min followed by 5 wash steps using PBS. The amount of antibodies and protein introduced into the NP was determined by Coomassie Plus Protein Assay Reagent (Pierce, Landsmeer, The Netherlands).

### 4.5. Dynamic Light Scattering, Zeta-Potential and Transmission Electron Microscopy

The NPs were characterized for average size, polydispersity index and surface charge (zeta-potential) by dynamic light scattering (DLS). Briefly, 50 µg of NP sample in 1 mL of ultrapure MilliQ H_2_O were measured for size using a Zetasizer (Nano ZS, Malvern, UK).

Transmission Electron Microscopy (TEM). Drops of NP was deposited over carbon-coated formvar films on copper grids and stained with a conventional negative staining for TEM using 2% uranyl acetate. The sample was analysed with a transmission electron microscope (JEOL JEM 1010 (Jeol, Akishima Tokyo, Japan) at an accelerating voltage of 80 kV. The images were obtained with a CCD Megaview III (SIS) camera (Münster, Germany).

### 4.6. Flow Cytometry Analysis of Binding and Uptake of the Various NP Vaccines by DCs

Binding and uptake of uncoated or ligand-coated fluorescent NP vaccine harbouring FITC-TT peptide by human DCs was studied using flow cytometry. NP vaccine (20 μg NP/mL) were incubated with 10^5^ DCs for 1 h in X-VIVO 15 medium supplemented with 2% human serum at 4 or 37 °C to determine binding and uptake, respectively. Next, cells were washed, and mean cell fluorescence determined by flow cytometry using a FACS Calibur (BD Biosciences, Franklin Lakes, United States). In part of the experiments, a mixture of the anti-human DC-SIGN antibodies AZN-D1 (50 µg/mL) and AZN-D2 (50 µg/mL), 50 µL FcR blocking reagent (Miltenyi Biotec, Bergisch Gladbach, Germany) or a combination of both were added to the cells 30 min before adding the NP vaccine to pre-block DC-SIGN and FcRs. The cells were analysed by flow cytometry using a FACS Calibur TM system.

### 4.7. DC Maturation and Activation

Human DCs (10^5^ cells/well) were plated (96 well plate) and incubated in the presence or absence of soluble or NP-encapsulated poly I:C (0.19 µg/mL), R848 (0.04 µg/mL) and TT-peptide Ag (0.2 µg/mL) for 48 h. NP vaccine carriers were uncoated or coated with biotinylated DC-SIGN ligands. After 2 days, supernatants were harvested and cytokine levels were determined by fluorescent bead immunoassay (flow cytometry human Th1/Th2 11plex kit; Bender Med Systems GmbH, Vienna, Austria). DCs were isolated and stained for maturation markers CD80, CD83, CD86 and CCR7 and analysed by flow cytometry. Relative expression levels were determined by dividing mean fluorescent intensities of experimental samples by those of untreated DCs.

### 4.8. Human CD8+ T Cell Activation

DCs (10^5^ cells/well) were incubated with 1.5 µg/mL gp100 (272–300) peptide, 1.0 µg/mL poly I:C and 0.32 µg/mL R848, either in soluble form or encapsulated in ligand-coated or uncoated NP vaccine (50 µg NP/mL). As a negative control, DCs were incubated with TLRLs and pulsed with irrelevant peptide Ag (GQAEPDRAHYNIVTFCCKCDSTLRLCV). Subsequently, DCs were co-cultured with TCR transfected, gp100 (280–288)-specific CD8+ T cells at a ratio of 1:10. After 1 day, cytokine production was measured in supernatant by cytometric bead array (Th1/Th2 kit, BD Biosciences) and CD69 expression by the T cells was determined by flow cytometry. Percentage of CD69+ cells in the total T cell population was plotted for experimental samples after correcting for percentage of CD69+ cells present obtained by pulsing with irrelevant peptide.

### 4.9. Human CD4+ T Cell Activation

The DCs and PBLs from venous blood obtained from healthy donors that responded to TT-Ag, using Ficoll gradient separation after informed consent was obtained. The DCs were incubated for 2 h with 30 and 60 ng/mL of FITC-TT peptide free or encapsulated within vaccine carriers coated with various ligands. In a different set of experiments, DCs were incubated with 30 and 60 ng/mL of FITC-TT peptide, poly I:C and R848 in soluble form or encapsulated within nanovaccine. Subsequently, DCs were washed and co cultured with TT-responsive PBLs (1:10 DC/T-cell ratio). Cytokine production was measured in supernatants after 24 h using fluorescent bead immunoassay (FlowCytomix human Th1/Th2 11plex kit; Bender MedSystems GmbH). Proliferative responses were determined after culturing DCs and T cells for 4 days by adding tritiated thymidine (1 OCi [0.037 MBq]/well; MP Biomedicals, Amsterdam, The Netherlands) to the cell cultures. Tritiated thymidine incorporation was measured after 16 h in a scintillation counter.

### 4.10. Statistics

Data were analysed by one-way or a two-way ANOVA followed by Bonferroni post-test.

## Figures and Tables

**Figure 1 molecules-24-01825-f001:**
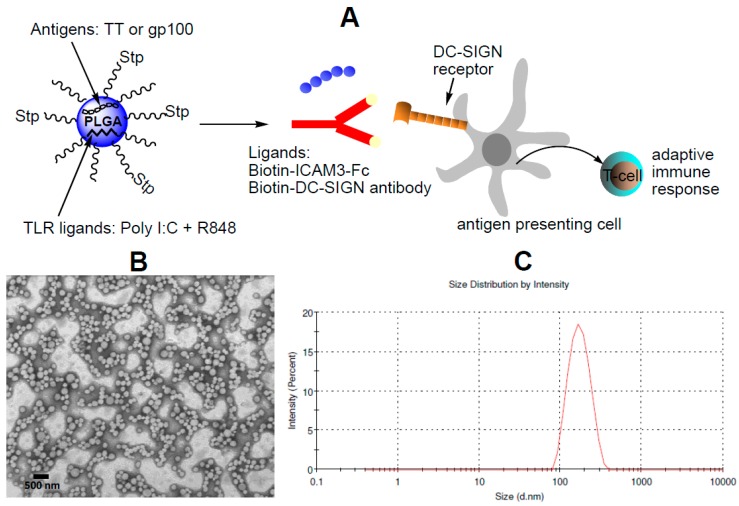
Schematic diagram of NP vaccine targeting DC-SIGN on human DCs. NP vaccine were generated carrying MHC class II-restricted TT peptide Ag or MHC class I - restricted gp100 peptide Ag in combination with the TLR3 ligand poly I:C and the TLR7/8 ligand R848. Carriers were coated with a lipid-PEG layer, to which streptavidin (Stp) was covalently attached. Biotinylated DC-SIGN ligands were introduced by binding to streptavidin (**A**). Shape and homogeneity of the NPs were observed by TEM (**B**). Diameter of the PLGA-NPs, as determined by dynamic light scattering (DLS) (**C**).

**Figure 2 molecules-24-01825-f002:**
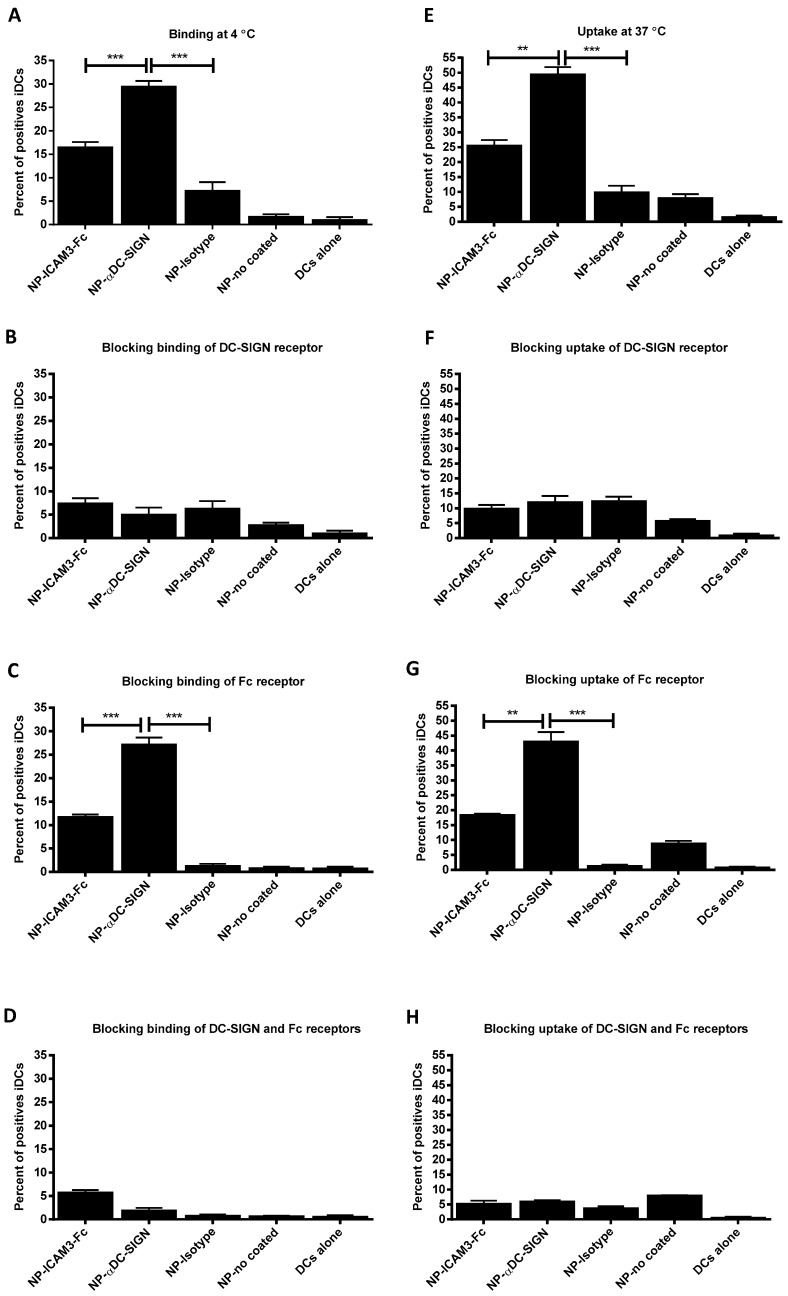
DC-SIGN specific antibodies are superior in targeting NP vaccine to DCs with respect to ICAM3-Fc. Binding (**A**–**D**) and uptake (**E**–**H**) of NP vaccine coated with various DC-SIGN ligands by DCs was evaluated. NP vaccine harbouring FITC-labelled TT peptide Ag and surface coated with ICAM3-Fc protein, DC-SIGN-specific Ab, isotype control Abs or without surface coat (no coated) were incubated with DCs for 1 h at 4 °C (**A**–**D**) or 37 °C (**E**–**H**). DCs cultured in medium without NP vaccines were included as a negative control. The percentage of DCs that had internalized the fluorescent carriers was determined by flow cytometry. To determine the specificity of vaccine carrier binding and uptake, experiments were performed without (**A**,**E**) and with pre-blocking the CRD of DC-SIGN (**B**,**F**), Fc receptors (**C**,**G**) or both (**D**,**H**). Data represent the mean value from 3 experiments performed in duplicate ± SD. Significant differences were analysed applying one way or two way ANOVA with Bonferroni post-tests, n.s. no significant, ** = *p* < 0.01 and *** = *p* < 0.001.

**Figure 3 molecules-24-01825-f003:**
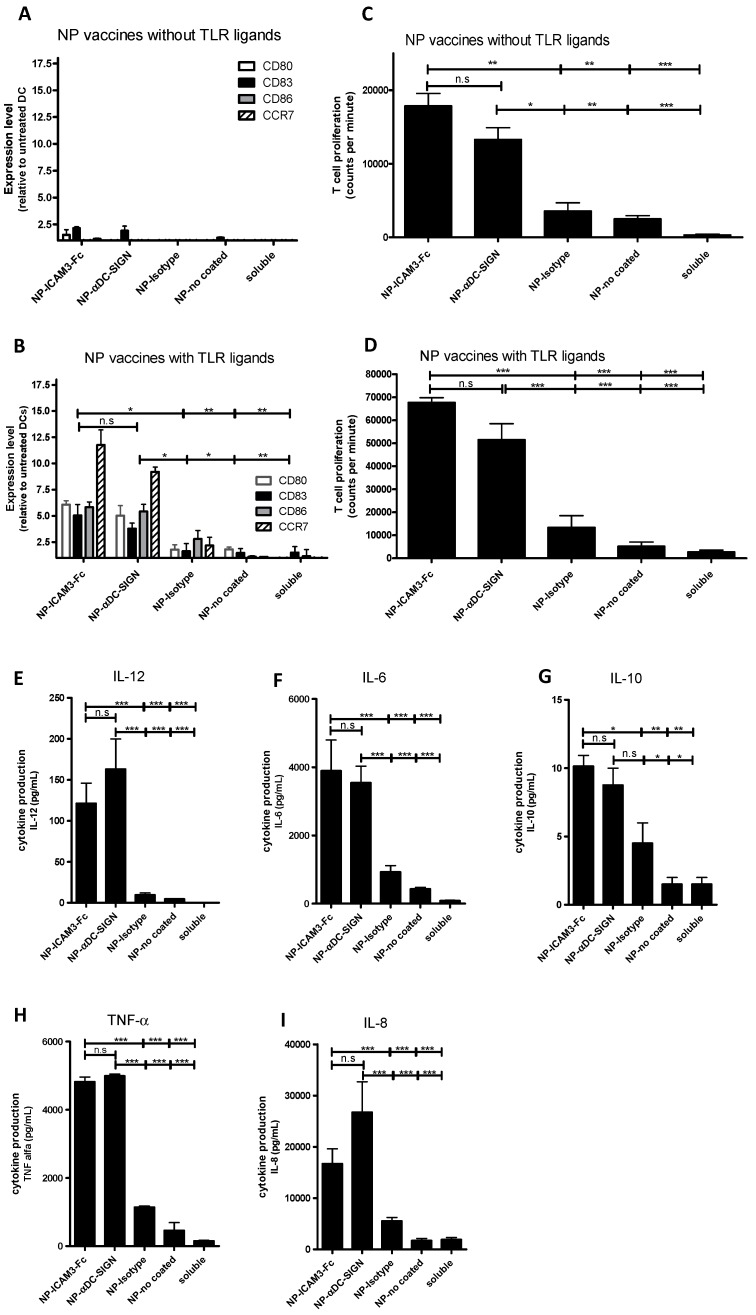
Targeted NP vaccines with TLR ligands induce DC maturation, cytokine production and T cell proliferation. The ability of the NP vaccine carrying peptide Ag to induce DC maturation was determined by analysing the induction of DC surface markers CD80, CD83, CD86 and CCR7 by flow cytometry. (**A**) DCs were cultured with uncoated (no) and ligand-coated NP vaccine harbouring peptide Ag but no TLRLs. (**B**) DCs were cultured with uncoated and ligand-coated NP vaccine harbouring peptide Ag and TLRLs. Untreated DCs and DCs cultured with soluble peptide Ag in the presence or absence of soluble TLRLs (soluble) served as controls. Relative expression levels were determined by dividing mean fluorescent intensities of experimental samples by those of untreated DCs. Three independent experiments were performed using DCs from different donors, showing similar results. Data represent mean expression levels of one representative experiment performed in triplicate. (**C**,**D**). Human monocyte-derived DCs were cultured in the presence of uncoated (no) and ligand-coated NP vaccine. As a control, DCs were cultured with soluble TT peptide Ag without NP vaccine carriers (soluble). NP vaccine contained (**C**) TT peptide or (**D**) TT peptide and TLRLs. Subsequently, autologous TT-responsive PBLs were added. T cell proliferation was measured by tritium thymidine incorporation assay. Data represent means ±SD of four independent experiments. Human monocyte-derived DCs were cultured in the presence of uncoated and ligand-coated NP vaccine harbouring peptide Ag and 0.19 µg/mL poly I:C and 0.04 µg/mL R848) for 48 h and supernatants were harvested to determine cytokine production levels (**E**) IL-12; (**F**) IL-6; (**G**) IL-10; (**H**) TNF-α; (**I**) IL-8. As a control, DCs were cultured at equal concentrations of soluble, non-encapsulated peptide Ag (0.2 µg/mL), poly I:C (0.19 µg/mL) and R848 (0.04 µg/mL). Two independent experiments were performed using DCs from different donors. Data represent mean cytokine expression levels of one experiment performed in duplicate. Significant difference were analysed applying one way or two way ANOVA with Bonferroni post-tests, n.s. no significant, * = *p* < 0.05, ** = *p* < 0.01 and *** = *p* < 0.001.

**Figure 4 molecules-24-01825-f004:**
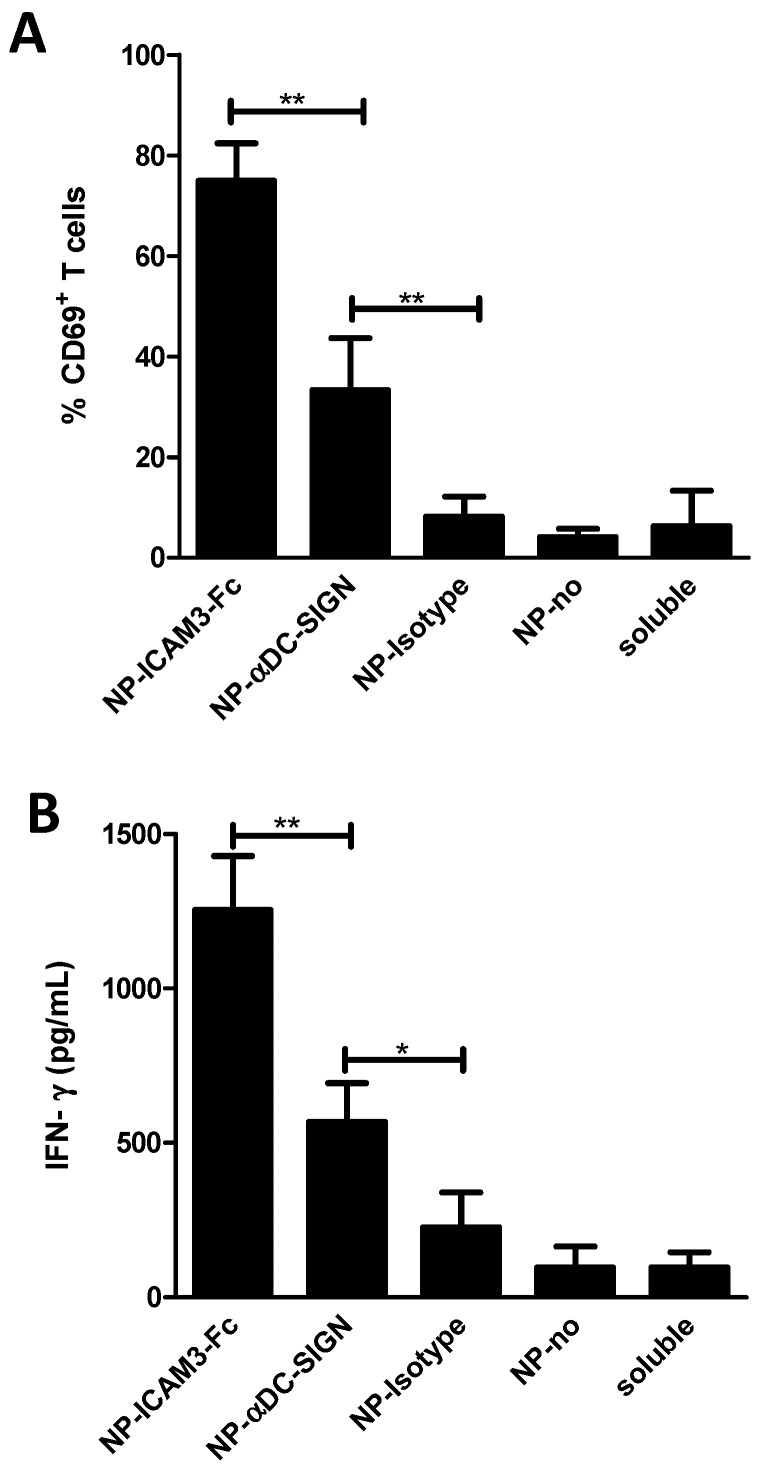
ICAM3-Fc coating of NP vaccines induces stronger CD8^+^ T cell activation and IFN-Ɣ production than antibody coating. Human monocyte-derived DCs were cultured in the presence of uncoated (no) and ligand-coated NP vaccine harbouring gp100_272-300_ peptide Ag and TLRLs. Untreated DCs and DCs cultured with soluble gp100_272-300_ peptide and TLRLs (soluble) served as controls. Next, DCs were co-cultured with gp100_280-88_-specific naïve CD8^+^ T cells. (**A**) CD8^+^ T cell activation was determined by measuring expression of the early T cell activation marker CD69. Data represent mean values ±SD of 5 experiments performed in duplicate. (**B**) CD8^+^ T cell function was determined by measuring IFN-γ levels in the supernatant. Data are means ±SD of 3 experiments performed in duplicate. Significant difference were analysed applying one way or two way ANOVA with Bonferroni post-tests, n.s. no significant, * = *p* < 0.05, ** = *p* < 0.01.

**Figure 5 molecules-24-01825-f005:**
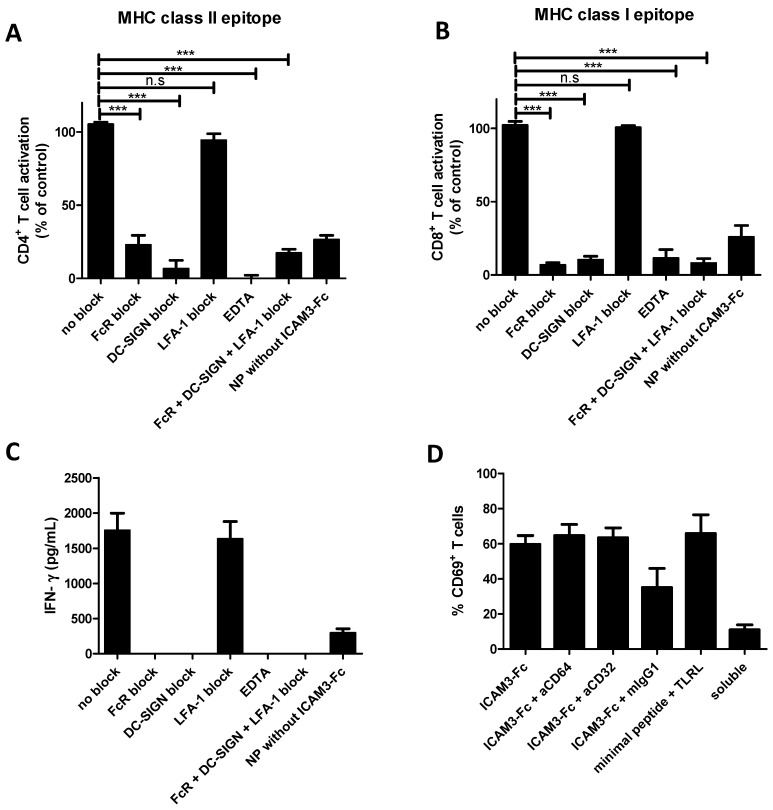
Antigen presentation by DCs following ICAM3-Fc targeted NP vaccines requires DC-SIGN binding and is blocked by excess IgG. (**A**) DCs were cultured in the presence of NP vaccines carrying the MHC class-II restricted specific TT peptide. Uncoated NP vaccine (NP without ICAM3-Fc), ICAM3-Fc-coated NP vaccine (no block, positive control) or a combination of ICAM3-Fc-coated NP vaccine and excess human IgG (FcR block), DC-SIGN-specific antibodies (DC-SIGN block), LFA-1 blocking antibody (LFA-1 block), EDTA or FcR block + DC-SIGN block + LFA-1 block (FcR + DC-SIGN + LFA-1 block). Subsequently, DCs were cultured with TT-responsive PBLs and T cell proliferation was measured by tritium thymidine incorporation assay. T cell proliferation is given as a percentage of T cell proliferation of the positive control (no block). Specific T cells. Data represent mean values ±SD of 3 experiments performed in triplicate, using cell from unrelated donors. (**B**,**C**) DCs were cultured in the presence of NP vaccines carrying the MHC class-I restricted gp100 (272–300) peptide. Uncoated NP vaccine (NPs without ICAM3-Fc), ICAM3-Fc-coated NP vaccine (positive control) or a combination of ICAM3-Fc-coated NP vaccine and excess human IgG (FcR block), DC-SIGN-specific antibodies (DC-SIGN block), LFA-1 blocking antibody (LFA-1 block), EDTA or FcR block + DC-SIGN block + LFA-1 block (FcR + DC-SIGN + LFA-1 block). Subsequently, DCs were cultured with gp100 (280–288-specific naïve CD8^+^ T cells). Data represent mean values ±SD of 3 experiments performed in triplicate, using cell from unrelated donors. Untreated DCs and DCs cultured with soluble gp100 (272–300) peptide and TLRLs (soluble) served as controls. Next, DCs were co-cultured with gp100 (280–288-specific naïve CD8^+^ T cells). T cell activation was determined either by measuring CD69 expression levels and given as a percentage of the positive control (no block) (**B**) or by measuring IFN-γ levels in the supernatant (**C**). Data are means ±SD of 3 experiments performed in duplicate using cells of unrelated donors. (**D**) DCs were cultured in the presence of NP vaccines carrying the MHC class-I gp100 (272–300) peptide Ag and TLRLs coated with ICAM3-Fc, either in the presence or absence of 20 µg/mL CD64 or CD32 blocking antibodies or control mIgG1. TLRL matured DCs cultured in the presence of soluble gp100 (272–300) peptide served as positive controls or pulsed with excess gp100 (280–288) peptide without activation as negative. Subsequently, DCs were co-cultured with gp100 280–288-specific naïve CD8^+^ T cells. T cell activation was determined by measuring CD69 expression levels. Data represent mean values ±SD of 3 experiments performed in triplicate, using cell from unrelated donors. Significant difference were analysed applying one way or two way ANOVA with Bonferroni post-tests, n.s. no significant, *** = *p* < 0.001.

**Figure 6 molecules-24-01825-f006:**
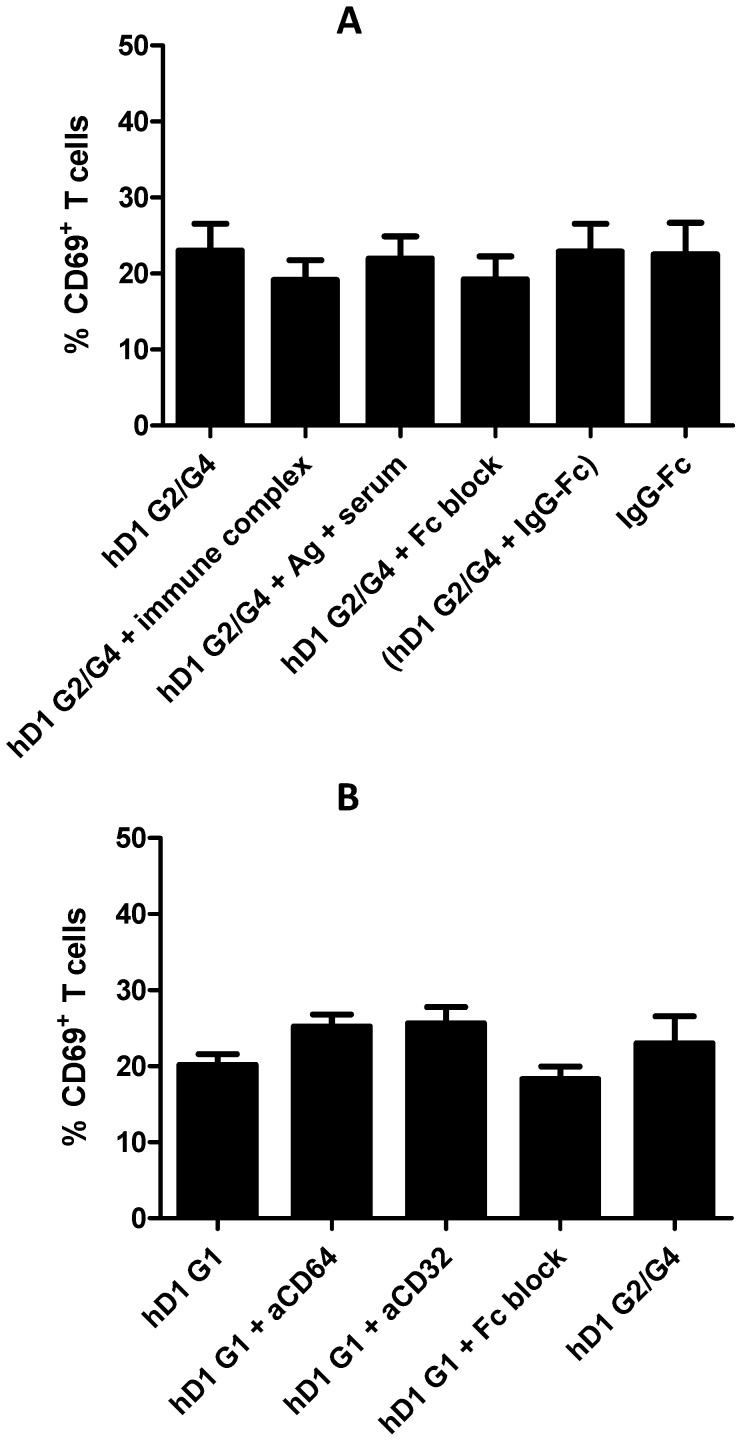
Combining DC-SIGN and FcR binding is not sufficient to boost antigen presentation. Human monocyte-derived DCs were cultured in the presence NP vaccines harbouring gp100 (272–300) peptide Ag and TLRLs. Subsequently, DCs were co-cultured with gp100 280-288-specific naïve CD8^+^ T cells. The percentage of activated T cells is determined by measuring CD69 expression. (**A**) NP vaccines were coated with DC-SIGN targeting Ab hD1 G2/G4. Immune complex was added to some samples to trigger FcRs (immune complex), whereas samples with irrelevant Ag and non-reactive serum served as negative control (Ag + serum). Other samples contained NP vaccines coated with hD1 G2/G4 in the presence of excess IgG (Fc block), NP vaccines coated with a combination of functional Fc domains and hD1 G2/G4 (hD1 G2/G4 + IgG-Fc) or NP vaccines coated only with functional Fc domains (IgG-Fc). (**B**) To block FcR binding of NP vaccines coated with hD1 G1, 20 µg/mL CD64 or CD32 blocking antibodies or excess IgG (Fc block) was added to the culture medium. (**C**) NP vaccines were coated with the DC-SIGN targeting Ab hD1 G1, which carries an unmodified, functional IgG1 Fc domain or with hD1 G2/G4. To analyse the role of FcR, it was blocked as in (**B**). Data are means ±SD of 3 experiments performed in duplicate using cells of unrelated donors.

**Table 1 molecules-24-01825-t001:** Determination of loading efficiency of Ag and TLR-Ls inside of NP vaccines. The efficiency of loading NP vaccines harbouring gp100:272–300 long peptide or FITC-TT peptide and TLR-Ls (R848 and Poly I:C) were determined by RP-HPLC and was also determined by nanodrop system.

NP Vaccines	R848 (mg/mg NP) (% w/w)	Poly I:C (mg/mg NP) (% w/w)	Ags (mg/mg NP) (% w/w)
NP-(FITC-TT)	------	------	25.9 ± 4.8 (64.8%)
NP-(gp100, R848, Poly I:C)	6.4 ± 2.3 (40%)	20.1 ± 3.1 (14.4%)	30.0 ± 5.2 (75%)
NP-(FITC-TT, R848, Poly I:C)	4.0 ± 1.2 (25%)	18.9 ± 4.2 (23.3%)	20.1 ± 2.9 (50.3%)

**Table 2 molecules-24-01825-t002:** Physicochemical characterization of NP vaccines. NP vaccines were characterized by DLS measurements, zeta potential measurements and quantification of surface ligands per NP. NP vaccine size data represent the mean value ± SD of ten readings from dynamic light scattering measurements. Zeta potential data represent the mean value ± SD of five readings. The amount of ligands bound to the NP surface was determined by Coomassie Plus Protein Assay Reagent (Pierce, Landsmeer, The Netherlands).

NP Vaccines with Distinct Targeting Moiety	Nanosphere Size ± S.D. (nm)	PDI	Width (nm)	Zeta Potential ± S.D. (mV)	Ligands (# Ligands/NP) (Experimental Values)
NP-none	205.1 ± 12.3	0.160	75.6	−15.76 ± 1.60	----
NP-ICAM3-Fc	234.1 ± 17.2	0.217	89.4	−16.12 ± 6.46	520
NP-αDC-SIGN	237.1 ± 31.9	0.213	112.7	−12.96 ± 1.57	390
NP-isotype (h5G1.1)	236.6 ± 27.2	0.187	92.5	−11.66 ± 2.34	401

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
