# Peer review of "ICAM3-Fc Outperforms Receptor-Specific Antibodies Targeted Nanoparticles to Dendritic Cells for Cross-Presentation"

_molecules, 2019, doi:10.3390/molecules24091825_

Round 1

Reviewer 1 Report

In this manscript Cruz et al. intend to optimize antigen presentation to dendritic cells in the context of vaccine-based anti-tumor therapy. In their approach they target NPs to dendritic cells by using the ICAM-3 ligand for cross-presentation. The manuscript contains several sets of data including NP characterizations and tests on human DCs and T-cells. Several experiments yield surprising results which are mostly be explaind by assumptions. I have several comments and questions:

* The authors study the effects on DC's and T-cell proliferation. However, I wonder how specific their chosen targetting moiety is for DC cells? ICAM3 binds to LFA-1, which is present on many different immune cells. Did the authors test any other cells? This is particularly important given the quest for cancer-vaccines which do no longer require in vitro Ag presentation to DCs.

* How pure are the monocyte-derived DCs after isolation? Are any other cells still present which could attribute for some of the observed results?

* The authors state in the discussion section that they have additional data showing that ICAM3-Fc coated NP vaccines were added to DCs and T cells from differen donors and no effects were observed. Since this observation strengthens their conclusion, these data should be shown (maybe as supplementary).

Author Response

Editor

Molecules

May  6, 2019

Dear Prof. Dr. Editor,

Included please find our revised version of the Manuscript ID: molecules-492028. Title: " ICAM3-Fc outperforms receptor-specific antibodies targeted nanoparticles to dendritic cells for cross-presentation" in which we addressed the concerns expressed by the reviewers. All changes to the manuscript have been highlighted for easy identification. We also include here a point-by-point response to address all of the reviewers’ comments.

We hope that you will find the revised manuscript acceptable for publication in your prestigious journal. Please do not hesitate to contact us if there are any additional aspects that require attention.

Thanking you in advance your kind attention, we are looking forward to hearing from you. With our best regards

Sincerely yours,

Dr. Luis J. Cruz

Asst. Prof. Dr. Luis J. Cruz

Translational Nanobiomaterials and Imaging, Department of Radiology, Bldg.1, C2-S.

Leiden University Medical Center, Albinusdreef 2

2333 ZA Leiden, The Netherlands

Tel: +31 71 5265764

Email: l.j.cruz_ricondo@lumc.nl

Reviewer: 1

Answers to the reviewer # 1

Comments and Suggestions for Authors

In this manuscript Cruz et al. intend to optimize antigen presentation to dendritic cells in the context of vaccine-based anti-tumor therapy. In their approach they target NPs to dendritic cells by using the ICAM-3 ligand for cross-presentation. The manuscript contains several sets of data including NP characterizations and tests on human DCs and T-cells. Several experiments yield surprising results which are mostly be explained by assumptions. I have several comments and questions:

* The authors study the effects on DC's and T-cell proliferation. However, I wonder how specific their chosen targeting moiety is for DC cells? ICAM3 binds to LFA-1, which is present on many different immune cells. Did the authors test any other cells? This is particularly important given the quest for cancer-vaccines which do no longer require in vitro Ag presentation to DCs.

The reviewer is right, but even ICAM3 may bind to other leukocytes and that LFA-1 is a receptor por ICAM3, we discovered that a novel DC-specific C-type lectin, DC-SIGN, binds ICAM-3 with high affinity. DC-SIGN, which is abundantly expressed by DC both in vitro and in vivo, mediates transient adhesion with T cells (Teunis B.H Geijtenbeek, Ruurd Torensma, Sandra J van Vliet, Gerard C.F van Duijnhoven, Gosse J Adema, Yvette van Kooyk, Carl G FigdorIdentification of DC-SIGN, a Novel Dendritic Cell–Specific ICAM-3 Receptor that Supports Primary Immune Responses. Cell, Vol. 100, 575–585, 2000). LFA-1 binds to ICAM-1 with high affinity, less strongly to ICAM-2, and only very weakly to ICAM-3 (Bleijs, D.A. et al. (1999) Costimulation of T cells results in distinct IL-10 and TNF-α cytokine profiles dependent on binding to ICAM-1, ICAM-2 or ICAM-3. Eur. J. Immunol. 29, 2248–2258). DC-SIGN binds with very high affinity to ICAM-2 as well as ICAM-3 (Geijtenbeek, T.B. et al. (2000) DC-SIGN–ICAM-2 interaction mediates dendritic cell trafficking. Nat. Immunol. 1, 353–357) (Diederik A. Bleijs, Teunis B.H. Geijtenbeek, Carl G. Figdor, Yvette van Kooyk. DC-SIGN and LFA-1: a battle for ligand. TRENDS in Immunology 2001, 22(8): 457)

* How pure are the monocyte-derived DCs after isolation? Are any other cells still present which could attribute for some of the observed results?

The yield of the standard procedure of purification and generation of dendritic cells, is not of 100%, but the other remaining cells than dendritic cells are a minority. Moreover, in our experience and by the general knowledge, it is impossible that other cells could influence or interfere to the results observed. Furthermore, the flow cytometry studies in the manuscript, only are expressed as positive cells, showed very pour contamination by cells different to dendritic cells.

* The authors state in the discussion section that they have additional data showing that ICAM3-Fc coated NP vaccines were added to DCs and T cells from different donors and no effects were observed. Since this observation strengthens their conclusion, these data should be shown (maybe as supplementary).

It has been done. Mixed lymphocyte culture is a technique to probe histocompatibility, as the two individuals (except monozygotic twins) are not genetically identical, different histocompatibility antigens on the surface of donor lymphocytes will activate the recipient lymphocytes and vice versa. The result is activation of DNA synthesis and the proliferation of lymphocytes. In our case, it was used as an unspecific test of lymphocyte proliferation, since as mentioned, unless monozygotic twins or immunodeficient patients, histocompatibility differences induces induce always immune response, but this was not enhanced by our ICAM3-Fc coated NP vaccines indicating ICAM3-Fc NP vaccines do not enhance CD8+ T cell activation by direct Ag-independent action, that is, it is not unspecific.

Reviewer 2 Report

The manuscript by Cruz LJ and associates studies different nanoparticle composotions specifically targeted to DC loading and uptake, and their effect on DC, T cell proliferation, and cross-presentation induced CD8+ T cell response. This is an important research for efficient vaccine design and development.

Specific comments:

Manuscript needs extensive editorial work.

Abstract. Why ABCAM-Fc fusion protein is mentioned as mimicking the natural ligand if it is a fusion protein containing a natural ligand?

Introduction. Reference statements presented in lines 50-54. Check discussion??? Lines 58-59: clarify for what purpose(s) PLGA NP use was approved by FDA.

Results. In figure legend for Figure 1 clearly state what is shown on panels A, B, and C.

Show statistics for Tables 1 and 2.

Line 416. Does human serum itself show any effect on DC maturation in cultures?

Line 418.Leucocytes are obtained from buffy coats, please clarify.

Line 429. ICAM3-Fc or DC-SIGN Ab, not both simultaneously.

Line 440. What kind of peptides were encapsulated in such way? If authors mean FITC-TT peptide, this procedure was already mentioned above (line 432). Be clear in composition description of all PLGA NPs. May be it is better to numerate the composition of each.

Line 445. Surface carboxyl groups of what?

Section 4.5. It is unclear what kind of information important for this research provides the described measurements of dynamic light scattering. Similar for Zeta potential.

Section 4.7. It is unclear how the human DC cultures were set up, what kind of plates and medium were used, what was a cell number used for their seeding. It is also unclear here why such different concentrations of NP-encapsulated peptides were used.

 Section 4.8. What served as an irrelevant peptide Ag?

Section 4.9. how the response to TT-Ag was evaluated in selected donors?

Figure 2. Figure title. What kind of Abs are superior? The iDC mark on Y axes is not clearly defined in Figure legend.

Line 180. Ab.

Figure 3. The selected Y axes scale on panel A does not allow us to see the differences between groups in marker expression levels. The results in this figure show no difference in T cell stimulation and cytokine production between DC stimulated with NP-ICAM3-Fc and NP-a-DC-SIGN Ab.

Author Response

Reviewer # 2

Answers to the reviewer # 2

Comments and Suggestions for Authors

The manuscript by Cruz LJ and associates studies different nanoparticle compositions specifically targeted to DC loading and uptake, and their effect on DC, T cell proliferation, and cross-presentation induced CD8+ T cell response. This is an important research for efficient vaccine design and development.

Specific comments:

Manuscript needs extensive editorial work.

Abstract. Why ABCAM-Fc fusion protein is mentioned as mimicking the natural ligand if it is a fusion protein containing a natural ligand?

The original characteristics of a protein is not always conserved in a fusion protein, being able to vary their properties, i.e., affinities. Sometimes structural changes are induced that may modify its functionality, but not in our case. However the statement is not critical and has been suppressed.

Introduction. Reference statements presented in lines 50-54. Check discussion??? Lines 58-59: clarify for what purpose(s) PLGA NP use was approved by FDA.

Paragraphs modified as suggested by the reviewer.

Results. In figure legend for Figure 1 clearly state what is shown on panels A, B, and C.

Description of each panel has been added.

Show statistics for Tables 1 and 2.

The statistic on the diameter and zeta potential is calculated authomaticaly by the software from the dynamic light scattering measurements.

Line 416. Does human serum itself show any effect on DC maturation in cultures?

Normally, the human serum itself show low effect on DC maturation in cell cultures. Moreover, we always used it as control in each experiment and subtract the value.

Line 418. Leucocytes are obtained from buffy coats, please clarify.

Leucocytes were obtained from venous blood obtained from healthy donors using Ficoll gradient separation.

Line 429. ICAM3-Fc or DC-SIGN Ab, not both simultaneously.

Reviewer is right. We have changed and by or.

Line 440. What kind of peptides were encapsulated in such way? If authors mean FITC-TT peptide, this procedure was already mentioned above (line 432). Be clear in composition description of all PLGA NPs. May be it is better to numerate the composition of each.

Done

Line 445. Surface carboxyl groups of what?

Done. Surface carboxyl groups of PLGA-PEG-COOH 

Section 4.5. It is unclear what kind of information important for this research provides the described measurements of dynamic light scattering. Similar for Zeta potential.

The text has been rephrased accordingly.

Section 4.7. It is unclear how the human DC cultures were set up, what kind of plates and medium were used, what was a cell number used for their seeding. It is also unclear here why such different concentrations of NP-encapsulated peptides were used.

Done.  

Section 4.8. What served as an irrelevant peptide Ag?

Done in section 4.8.. We use as a irrelevant peptide Ag (GQAEPDRAHYNIVTFCCKCDSTLRLCV).

Section 4.9. How the response to TT-Ag was evaluated in selected donors?

It must be clarified in the manuscript

TT proliferation is a standard test to measure if the donor of the leucocytes has been encountered TT in the past.

Figure 2. Figure title. What kind of Abs are superior? The iDC mark on Y axes is not clearly defined in Figure legend.

Done. DC-SIGN specific antibodies have been added in the title of the figure 2. It was DC and iDC.

Line 180. Ab.

It has been corrected

Figure 3. The selected Y axes scale on panel A does not allow us to see the differences between groups in marker expression levels. The results in this figure show no difference in T cell stimulation and cytokine production between DC stimulated with NP-ICAM3-Fc and NP-a-DC-SIGN Ab.

The scale is the same in panel A and B to be able to compare, but in panel A almost no induction of markers of activation are present by the absence of TLR. The no differences in T cell stimulation that is mentioned by the reviewer is clearly stated in section 2.3.

The manuscript is amended to address the comments and changes are highlighted in green for easy identification.

Round 2

Reviewer 1 Report

Dear authors,

Thank you for the answers to my comments. My only recommendation would be to include the clarifications on the receptor affinities into the manuscript.

Author Response

Done. This sentence below has been incorporated into the Manuscript. Line 116

ICAM-3 can also bind to LFA. However, LFA has to be activated to allow binding. A resting T cell expresses low affinity LFA and upon activation LFA changes its conformation and forms clusters on the membrane providing high affinity to ICAMs